# Identification and Characterization of *Arbutus unedo* L. Endophytic Bacteria Isolated from Wild and Cultivated Trees for the Biological Control of *Phytophthora cinnamomi*

**DOI:** 10.3390/plants10081569

**Published:** 2021-07-30

**Authors:** João Martins, Aitana Ares, Vinicius Casais, Joana Costa, Jorge Canhoto

**Affiliations:** 1Department of Life Sciences, Centre for Functional Ecology, University of Coimbra, Calçada Martim de Freitas, 3000-456 Coimbra, Portugal; joao.martins@uc.pt (J.M.); bioaitana26@gmail.com (A.A.); viniciuscasais@gmail.com (V.C.); jcosta@uc.pt (J.C.); 2Laboratory for Phytopathology, Instituto Pedro Nunes, 3030-199 Coimbra, Portugal

**Keywords:** *Arbutus unedo*, *Bacillus cereus*, bacterial endophytes, biological control, *Phytophthora cinnamomi*

## Abstract

*Arbutus unedo* L. is a resilient tree with a circum-Mediterranean distribution. Besides its ecological relevance, it is vital for local economies as a fruit tree. Several microorganisms are responsible for strawberry tree diseases, leading to production constrictions. Thus, the development of alternative plant protection strategies is necessary, such as bacterial endophytes, which may increase their host’s overall fitness and productivity. As agricultural practices are a driving factor of microbiota, this paper aimed to isolate, identify and characterize endophytic bacteria from strawberry tree leaves from plants growing spontaneously in a natural environment as well as from plants growing in orchards. A total of 62 endophytes were isolated from leaves and identified as *Bacillus*, *Paenibacillus*, *Pseudomonas*, *Sphingomonas* and *Staphylococcus*. Although a slightly higher number of species was found in wild plants, no differences in terms of diversity indexes were found. Sixteen isolates were tested in vitro for their antagonistic effect against *A. unedo* mycopathogens. *B. cereus* was the most effective antagonist causing a growth reduction of 20% in *Glomerella cingulata* and 40% in *Phytophthora cinnamomi* and *Mycosphaerella aurantia*. Several endophytic isolates also exhibited plant growth-promoting potential. This study provides insights into the diversity of endophytic bacteria in *A. unedo* leaves and their potential role as growth promoters and pathogen antagonists.

## 1. Introduction

*Arbutus unedo* L. is a small tree native in the Mediterranean basin as well as in Portugal and in South Ireland [1]. Commonly known as the strawberry tree, this member of the Ericaceae thrives on rocky marginal soils and dry lands and can tolerate a wide range of temperatures [2]. It is very important in southern European forest ecosystems, due to its ability to resprout after forest fires, a common scenario during the hot dry season. Besides avoiding erosion and desertification of extensive areas, it also prevents the colonization by invasive species [3]. Furthermore, it is very important from an economic point of view and its edible berries are the most important source of revenue for producers. Although they can be eaten raw, fruits are usually used in the production of a high-value alcoholic distillate [4]. The species is also a source of several bioactive compounds with application in the cosmetic and pharmaceutical industries [5]. Other applications are honey production and floriculture.

Due to the increasing demand for such products and a consequent market value growth, the production area has been rising considerably in recent years, although facing constraints due to phytosanitary problems. Several microorganisms, such as *Glomerella cingulata* [6], *Mycosphaerella* spp. [7] and *Phytophthora cinnamomi* [8], are known to be strawberry tree pathogens causing several diseases and leading to a reduction in plant fitness and constraints on crop production. The most aggressive is *P. cinnamomi*, an invasive and widespread oomycete, responsible for huge losses in agriculture and forestry worldwide, as it leads to tree decay [9], calling for adequate responses to improve agricultural practices and upkeep production. 

To achieve this goal, a deep understanding of plant defence mechanisms is required as well as the selection and breeding of improved genotypes. Still, a change in paradigm is needed, and an integrative approach must be followed taking into consideration all the different aspects influencing plant performance and productivity. In this sense, the microbiota should be taken into account, as they have great significance in natural and agricultural ecosystems [10,11]. In particular, endophytic bacteria may confer advantages to host plants, promoting their overall fitness and productivity by enhancing plant resistance to biotic and abiotic stress conditions, such as pathogen attacks and drought [10]. These microorganisms are known to be the dominant group in these communities and have been isolated from numerous plant hosts where they live in symbiosis [12,13]. Such beneficial effects are accomplished through the production of phytohormones, secondary metabolites with important bioactive activities (e.g., antifungal and antiviral) and by promoting an increase of mineral uptake or nitrogen fixation [12,13]. Such features might be important in plant breeding and can be accomplished through the integration of specific endophytic bacteria genes on plants or by the inoculation of bacterial strains into the plant [10]. In addition, to enhance crop output, this approach can be a sustainable alternative for conventional pesticides and fertilizers, with economic and environmental benefits [14]. The structural characterization of plant microbiota, from their composition to the intricate interaction with the host plant, is essential for this knowledge to be applied by breeders and farmers [15]. Although the endophyte population dynamics have not yet been fully understood, several factors have been reported to cause variations among endophyte populations [13]. In particular, the surrounding environment and agricultural practices are considered the most crucial driving factors shaping the microbiota composition and functionality [16].

For this reason, the aim of this work was the isolation and molecular identification of endophytic bacteria from strawberry tree leaves from plants growing spontaneously in a natural environment as well as from plants growing in a production orchard, in order to compare the bacterial diversity between the two groups and evaluate the possible effects of agricultural practices on strawberry tree bacterial endophytes. Moreover, to assess the function of these bacterial endophytes in plant defence, its antagonism effect was tested against the most important strawberry tree pathogens, namely, *Glomerella cingulata*, *Mycosphaerella aurantia* and *Phytophthora cinnamomi*. Insights about the bacteria endophytic diversity and their plant growth-promoting features and antagonism effects are provided and the implications of these results on biological control strategies, plant phenotyping and breeding are discussed.

## 2. Results

### 2.1. Isolation of Endophytic Bacteria from Cultivated and Wild Strawberry Trees

A total of 62 endophytic strains were isolated from the leaves of strawberry tree. A similar number of isolates were obtained on the two media (ABM2 and 1/10 869) and three temperatures tested (20, 25 and 30 °C). In contrast, the number of isolates obtained greatly varied according to the medium pH (5.5, 7.0 and 8.5). A considerably higher number of isolates was obtained for pH 7 on both mediums and no bacteria growth was observed on the ABM2 medium with pH 5.5 (Appendix A).

From the 62 isolates, 32 were obtained from wild plants, whereas the other 29 from the cultivated ones. Random Amplified Polymorphic DNA (RAPD) profiles were obtained for all the 62 isolates and grouped in 50 different clusters. Most of the RAPD profiles (39) were unique. Eight clusters were formed by two isolates while the other three by three or more isolates. Isolates selected for 16S rRNA gene amplification and sequencing were identified belonging to five genera (*Bacillus*, *Paenibacillus*, *Pseudomonas*, *Sphingomonas* and *Staphylococcus*) and 17 bacteria species (Figure 1, Appendix A).

A total of 21 distinct RAPD profiles were obtained from the bacterial isolates from cultivated plants, while 27 were obtained from strains isolated in wild plants (Figure 2a and Figure 3). Only two RAPD profiles were common between isolates from cultivated and wild plants, identified as *B. toyonensis* and *P. humicus*. This difference was more marked in the number of species since seven different species were restricted to wild plants, while only four species were exclusively found in cultivated plants (Figure 2b and Figure 3). Additionally, five bacterial species were shared between wild and cultivated plants, namely, *B. cereus*, *B. toyonensis*, *P. humicus*, *P. taichungensis* and *S. epidermidis*. In terms of distribution between replicates, *P. taichungensis* was found on all the three replicates, from cultivated and isolated plants, whereas *B. cereus* was identified in all replicates from cultivated plants and *P. humicus* in wild plants (Figure 2c–d).

*Bacillus* and *Paenibacillus* were the genera more often recovered, comprising 17 and 26 RAPD profiles, corresponding to 19 and 32 isolates, respectively. On the contrary, only two *Pseudomonas* species (*P. avellanae* in cultivated and *P. poae* in wild trees) were identified corresponding to three RAPD profiles with six isolates (Appendix A). The same number of *Staphylococcus* species were identified (*S. epidermidis* and *S. capitis*), with three RAPD profiles and three isolates. Two isolates obtained from cultivated plants, with the same RAPD profile, were identified as *Sphingomonas hankookensis*, the only species found from this genus. Although seven species had only one RAPD profile, others had a considerably higher number of RAPD profiles: *B. cereus* (8), *P. taichungensis* (12) and *P. humicus* (15). Although the other endophytes isolated from strawberry tree leaves have already been identified in other plant species, as far as we know, this is the first report of *Paenibacillus etheri* as a plant endophyte (Table 1).

The Simpson diversity index (D) calculated for both populations was 0.11 ± 0.02 for cultivated plants and 0.13 ± 0.07 for the wild ones, whereas the Shannon index (H’) was 2.22 ± 0.16 in the cultivated group and 2.19 ± 0.70 in the wild one. Finally, Pielou’s evenness index (J’) was 1 ± 0 in the cultivated plants and 1 ± 0 in the wild population (Figure 3). No statistical differences were observed between groups for all the diversity indexes calculated.

### 2.2. Antagonism Effect of Endophytic Bacteria against Plant Pathogens

Sixteen strains were tested for the ability to inhibit the growth of strawberry tree pathogens. Although most of the isolates caused a reduction in the growth (inhibition) of *Glomerella cingulata* after one week in culture, this antagonism effect was reduced over time (Figure 4a–c). After 3 weeks in culture, the isolate Au06 (*B. cereus*) proved to be the most effective antagonist, causing a reduction in growth close to 30%. Several of the tested bacterial isolates had the opposite effect, promoting the growth of *G. cingulata* (between 10% and 30%): Au3 (*P. humicus*), Au15 (*P. pabuli*), Au 21 (*B. simplex*), Au 47 (*P. etheri*), Au 53 (*P. taichungensis*) and Au 61 (*B. taxi*) (Figure 4c). 

In the case of *P. cinnamomi*, a slight reduction in growth (less than 20%) was observed after one week for most isolates (Figure 4d). This antagonist effect was intensified after two weeks (Figure 2e), but after three weeks in culture, the effect of the bacteria was barely noted, except for isolate Au06 (*B. cereus*), which caused a reduction in the growth of more than 40% (Figure 4f). 

Most of the bacterial isolates caused a reduction in the growth of *M. aurantia* after three and six weeks in culture (Figure 4g,h). After nine weeks, all the isolates proved to have an antagonist effect and caused a reduction in *M. aurantia* growth in some cases close to 40% (Figure 4i). 

Overall, *B. cereus* (Au06) was the most effective antagonist of *A. unedo* fungal pathogens tested in this study, as it was able to control their growth. It is important to note that the antagonism effect of *B. cereus* was caused at a distance and not by direct contact between microorganisms (Figure 5).

### 2.3. Plant Growth-Promoting Potential of Endophytic Bacteria Isolated from Strawberry Tree

From the 16 isolates tested, five were found to produce siderophores: Au01 (*P. avellanae*), Au06 (*B. cereus*), Au14 (*B. safensis*), Au39 (*P. poae*) and Au43 (*S. panni*); and four were able to solubilize phosphate: Au04 (*B. megaterium*), Au32 (*P. marchantiophytorum*), Au39 (*P. poae*) and Au61 (*Bacillus taxi*) (Table 2). Whereas most of the isolates tested were able to produce ammonia, namely, Au01 (*P. avellanae*), Au02 (*B. toyonensis*), Au04 (*B. megaterium*), Au05 (*B. toyonensis*), Au06 (*B. cereus*), Au08 (*B. toyonensis*), Au14 (*B. safensis*), Au21 (*B. simplex*) and Au39 (*P. poae*) (Table 2), the production of IAA was only detected in six bacterial species: Au01 (*P. avellanae*), Au04 (*B. megaterium*), Au21 (*B. simplex*), Au39 (*P. poae*), Au43 (*S. panni*) and Au61 (*B. taxi*) (Table 2). IAA quantification revealed the production of this hormone by all the isolates, but considerable differences in the amount of IAA between the bacterial species were found, with values ranging from 0.56 ± 0.43 for isolate Au32 (*P. marchantiophytorum*) and 10.98 ± 2.44 produced for Au01 (*P. avellanae*) (Table 2).

## 3. Discussion

The two different media used in this study proved to be efficient for the isolation of strawberry tree endophytic bacteria. Although the ABM2 medium is more complex when compared to 869, both had a common basal composition that was suitable for the recovery of bacteria. The difference observed in the number of isolates according to the pH of the medium was expected due to the influence of pH on shaping microbial communities. The higher number of isolates obtained at pH 7 indicates that most strawberry tree culturable endophytic bacteria are neutrophiles. 

A considerably higher number of RAPD profiles was obtained when compared to the number of species identified, which suggests a high inter-specific diversity. Another observation that reinforces this idea is the fact that no common RAPD profiles were found between replicates. The difference in the number of RAPD profiles and bacteria species between replicates on the cultivated plants suggests a more homogeneous microbiota within the population, whereas on the wild plants a higher heterogenicity seems to exist. Moreover, the total number of RAPD profiles as well as unique RAPD profiles were found to be higher in wild plants. The number of taxa identified in wild plants was also higher than in cultivated ones. Although these results suggest a higher diversity in the wild population when compared to the cultivated plants, diversity indexes were very similar between groups. Furthermore, Simpson and Pielou’s indexes suggest low dominance of species and evenness among populations. As mentioned before, agricultural practices might have great influence on microbiota structure. Nitrogen fertilization, for example, a common practice in modern intensive agriculture, proved to lead to a decline in plant microbiota diversity in maize [38]. Moreover, either to promote growth and/or to facilitate fruit collection, orchard plants are pruned to obtain a specific plant architecture, which may also influence microbiota structure [11]. Although cultivated plants analysed in this study are regularly watered and pruned, and the soil fertilized, these practices do not seem to greatly affect microbiota diversity when compared to wild populations.

The surrounding environment is another important source of microorganisms that might be transmitted horizontally and enter the plant through different paths integrating its microbiota [16]. Thus, surrounding plants may constitute a local reservoir of bacteria, and as plant diversity is hypothetically lower in an agricultural ecosystem when compared to a wild forest, this could lead to a decrease in plant microbiota diversity. Due to the lack of such niches in the cultivated population studied, the horizontal transmission between individuals might be compromised, whereas the microbiota diversity in the wild population might be promoted due to the existence of several other plant species. Still, the contribution of these bacteria reservoirs on plant microbiota structure and transmission of dominant bacteria species needs to be further evaluated, as such knowledge can be used to improve agricultural practices (e.g., the inclusion of ecological corridors to promote local diversity). 

Additionally, plant microbiota is also transmitted vertically by seeds or other propagation material. In this particular case, cultivated plants have been obtained by conventional seed germination. Thus, the origin of the seeds might be crucial to determine the microbiota composition of the plants. Nonetheless, future studies are required to evaluate the contribution of the propagation material to the plant core microbiome. 

*P. humicus* was found on most replicate samples while *P. taichungensis* was found on all the samples both from wild and cultivated plants. Because these plants are under different conditions, this result might indicate these two bacteria species are part of the strawberry tree core microbiota, while the other species constitute the satellite microbiota. However, further studies with more plant genotypes growing in different environmental conditions are yet to be conducted to confirm this result.

A fierce competition for space and nutrient resources occur in microbial communities, including between bacteria and fungi, through different mechanisms such as cell signalling and antibiotic production. These complex interactions allow the coexistence or exclusion of some species [39]. In this work, several bacteria were able to antagonize fungal pathogens (*G. cingulata* and *M. aurantia*) and an oomycete (*P. cinnamomi*). The most promising results were obtained with *B. cereus*, which was able to antagonize the three plant pathogens tested. Its antifungal effect has already been reported in the literature against *Fusarium oxysporum* f. sp. *lycopersici* [18]. However, to our knowledge, this is the first report of its antifungal effect against *G. cingulata*, *M. aurantia* and *P. cinnamomi*. Several *B. cereus* strains were isolated from both populations (cultivated and wild), which seems to be an indicator of its importance and prevalence among strawberry tree populations. The antifungal effect of other bacteria, such as *B. safensis*, against other plant pathogens (*Magnaporthe oryzae*) has also been reported in the literature [28]. However, no antagonism effect of *B. safensis* was observed in this work on the pathogens tested. 

Our characterization study also revealed the production of siderophores by five strawberry tree endophytic bacteria. The production of these low molecular weight compounds is an important competition mechanism, in particular under iron-limiting conditions [40]. Besides improving the nutritional status of bacteria, they also limit iron absorption by other organisms, such as fungi, which confers a competitive advantage and can be one of the mechanisms involved in biological control. Some of the isolates have also shown the ability to solubilize phosphate and produce ammonia, which are important bacterial features that enhances their nutritional status and have a significant effect on the competition with other microorganisms.

Bacteria might also produce IAA and other plant hormones. IAA production enhances microbial fitness as it stimulates cell wall loosening and nutrient leakage [39]. As one of the most important plant hormones, IAA is involved in several plant signalling pathways and greatly affects plant growth and development. Thus, the production of IAA by microorganism might be of great benefit to plants [41]. Although it is fairly known that most rhizobacteria can synthesize IAA, this is not the case for phyllosphere bacteria [42]. Nonetheless, our results revealed all the tested bacteria can produce IAA, even though in small quantities. For instance, the amount of IAA produced by *B. cereus* (1.72 µg mL^−1^) is much lower than that of 18 µg mL^−1^ quantified by Wagi and Ahmed [42]. This difference may be related to the experimental conditions (e.g., medium and temperature) or genetic factors. *B. megaterium*, *B. safensis* and *P. pabuli* have also been referred to in the literature as IAA-producing bacteria [20,23].

This characterization study provides important insights about strawberry tree endophytic bacteria physiological mechanisms and possible ecological interactions with important implications in future biotechnological applications. Overall, endophytic bacteria have a great influence on plant resilience against biotic and abiotic stresses and a huge impact on plant health and development, leading to an increase in agricultural production. Plant microbiota contributes to a reduction in greenhouse gas emissions and chemical inputs mostly due to its benefits on plant nutrition (e.g., nitrogen fixation) and potential to reduce plant disease incidence [11,16,43], leading to more sustainable agricultural practices. As pathogenic microorganisms are a chronic problem that affects food production, the dependency on agrochemicals is growing as agricultural production intensifies. Due to the negative impacts of agrochemicals (e.g., high cost, low efficiency, development of pathogen resistance and environmental impacts), a reduction in chemical inputs is a desirable goal for worldwide agriculture, making it urgent to develop alternative mitigation strategies. In this sense, biological control with bacteria can arise as an alternative, reliable crop protection method [40,44]. Furthermore, some bacteria species also have the potential for phytoremediation of contaminated soils [30].

Metabolomic studies, the identification of plant genes that are crucial for the microbial composition, as well as a better understanding of the microorganisms’ physiology and multitrophic interactions should be pursued in the near future, as they are key factors to elucidate the recruiting mechanisms and interactions between bacteria and host plants. Such knowledge would have great repercussions on plant phenotyping and breeding and would allow further development of plant protection strategies and forest management strategies [39,45]. Due to its importance on plant phenotype, microbiome populations should be taken into account on plant selection. Moreover, propagation and plant breeding may lead to an interaction disruption between the plant and its microbiota [10]. Thus, the selection of plant genotypes with an appetence to establish symbiotic relations with specific bacteria species and/or strains might be of high importance to maintain these beneficial interactions [46].

## 4. Materials and Methods

### 4.1. Collection of Plant Material

Plant material was collected during May 2018 from 30 *A. unedo* trees growing in the wild (40.043334, −7.904996) and 30 in an orchard (40.029581, −7.924739), in Pampilhosa da Serra, Coimbra district, central Portugal. The distance between the groups is approximately 2 km. Trees were 5 m apart on a 750 m^2^ area (50 m × 15 m). Three replicates from each location were used, each replicate consisting of 10 healthy adult trees (i.e., without any disease symptoms). Five leaves without any visible symptoms were randomly collected from each tree (a total of 50 leaves per replicate) and pooled together (Figure 6). Leaves were stored in sterile plastic bags at 4 °C and processed within 24 h. Plants from the wild location were growing under uncontrolled conditions, in a mixed forest with *Pinus pinaster* and shrubs (*Erica arborea*, *Calluna vulgaris* and *Ulex* spp.) as well as several weeds. In the orchard, an irrigation system was installed, plants were regularly pruned and fertilized, and the soil had been mobilized to remove weed species.

### 4.2. Isolation of Endophytic Bacteria

Leaves were sterilized as previously described by Eevers et al. [47]. Briefly, leaves were thoroughly washed in sterile ultra-pure water, surface-disinfected in ethanol (70%, *v*/*v*) for 90 s, sodium hypochlorite (1%, *w*/*v*) for 3 min and finally rinsed five times in sterile ultra-pure water. The last rinsing water was inoculated in alkaline buffered medium 2 (ABM2 [48]) and incubated at 25 °C for 72 h to confirm the efficiency of the process. Approximately 10 g of sterilized leaves were shredded in a blender with 10 mL PBS buffer (10 mM, pH 7.2), and filtered through a sterile gauze cloth to remove plant debris. Several serial dilutions were prepared (0, 10–1, 10–2, 10–3, 10–4) and 100 μL of each dilution was inoculated in Petri dishes containing ABM2 and 1/10 869 medium [49], at pH 5.5, 7.0 and 8.5, and cultured at three different temperatures: 20, 25 and 30 °C. Petri dishes were examined daily, and bacteria colonies were isolated in the same medium with the same pH and kept at 25 °C.

### 4.3. Identification of Endophytic Bacteria

#### 4.3.1. DNA Extraction and RAPD Fingerprint

For DNA extraction, a crude lysate was prepared by resuspending a bacteria colony on 50 μL of 0.5M NaOH, heated at 95 °C for 4 min and immediately chilled on ice for 10 min. After centrifugation at 10,000 g for 5 min to remove debris, the supernatant was collected and stored at −20 °C (adapted from Wiedmann-Al-Ahmad et al. [50]). RAPD-PCR was performed as previously described [51] to assess the genetic diversity of the bacterial strains using the primer OPA-03 (5′-AGTCAGCCAC-3′). PCR products were visualized on a 2 % (*w*/*v*) agarose gel stained with GreenSafe Premium (NZYTech, Oeiras, Portugal), under UV light. Clusters were formed by visual inspection based on the similarity and intensity of the fluorescence of each band observed in the RAPD profiles, namely, the number and weight of the bands when compared with each other and with the molecular weight marker. Strains from the same RAPD cluster were considered to be from the same species.

#### 4.3.2. Phylogenetic Analysis

Phylogenetic analyses were performed in the representative bacterial strains selected from the previously established clusters based on the RAPD-PCR fingerprinting analysis (1–2 strains per cluster). Isolates were identified based on the 16S rRNA gene sequence (approximately 1500 bp) amplified by polymerase chain reaction (PCR) using primers 27F (5′-AGAGTTTGATCCTGGCTCAG-3′) and 1525R (5′-GGTTACCTTGTTACGACTT-3′) [52]. PCR reactions contained 1 × NZYTaq II 2 × Colorless Master Mix (NZYTech), 10 mM primer, 50 ng DNA in a final reaction volume of 50 µL. The reactions were performed in a T100™ Thermal Cycler (BioRad, Hercules, CA, USA) under the following conditions: 95 °C for 5 min, followed by 30 cycles at 95 °C for 1 min, annealing at 55 °C for 1 min and extension at 72 °C for 2 min, before a final extension at 72 °C for 10 min. PCR products were visualized on a 1.5 % (*w*/*v*) agarose gel stained with GreenSafe Premium (NZYTech), under UV light. PCR products were purified and sequenced by StabVida (Oeiras, Portugal), using the reverse 519R primer (5′-GWATTACCGCGGCKGCTG-3′) [53] to obtain a partial sequence. Sequences were manually edited using Sequence Scanner Software 2 (Applied Biosystems, Foster City, CA, USA) and compared with the available data from the EZBioCloud databases version 2020.10.12 [54]. Those obtained from EZBioCloud with more than 97% similarity with our isolates were used for phylogenetic analysis and aligned using ClustalW [55] within MEGA (version 10.0 for Windows; [56]). This operation was done for each bacterial genus separately. Best suitable DNA substitution models were assessed using the “find best DNA/Protein Models (ML)” function on MEGA by implementing the Maximum Likelihood (ML) statistical method to test the goodness of fit to several models of evolution. According to the estimated values of all parameters for each model, the best-fitting model to the dataset from the 16S rRNA gene sequences was Kimura 2-parameter (K2) and gamma-distributed (+G) with invariant sites (+I) (=K2+G+I) [57] for *Bacillus* spp. and *Paenibacillus* spp., Hasegawa-Kishino-Yano (HKY+G+I) [58] for *Pseudomonas* spp., Tamura 3-parameter (T92+G+I) [59] for *Sphingomonas* spp. and Tamura-Nei (TN93+G+I) [60] for *Staphylococcus* spp. Phylogenetic reconstruction was calculated using the Maximum Composite Likelihood method with bootstrap values calculated from 1000 replicate runs. To confirm the identification, a larger sequence was obtained from 30 representative isolates using the forward primers 357F (5′-CCTACGGGAGGCAGCAG-3′) [61] and 803F (5′-ATTAGATACCCTGGTAGTC-3′) [53]. Similarly, sequences were compared with available data from EZBioCloud databases and a phylogenetic reconstruction was calculated using the Kimura 2-parameter model (K2+G+I) with the 16S rRNA sequences obtained from the *Arbutus unedo* 30 representative bacteria isolates along with the most similar sequences retrieved from the EZBioCloud database (Appendix A).

### 4.4. Antagonism Ability of Bacterial Endophytes against A. unedo Fungal Pathogens

Bacterial isolates were tested in vitro for their antagonistic effect against microorganisms (two fungi and one oomycete) known to be pathogenic to *A. unedo*, using the direct opposition method [62]. A strain randomly selected from each bacterial species from the genera *Bacillus*, *Paenibacillus*, *Pseudomonas* and *Sphingomonas* was used (a total of 16). Briefly, mycelia plugs with 5 mm taken from the outside ring of a 7-day-old actively growing colony of *G. cingulata*, *M. aurantia* and *P. cinnamomi*, from the Plant Biotechnology Laboratory collection (isolated from *A. unedo*), were placed at 1 cm from the wall of a Petri dish (90 mm) with PDA medium (DifcoTM Potato Dextrose Agar: 4 gL^−1^ potato starch, 20 gL^−1^ dextrose, 15 gL^−1^ agar). On the opposite side, 10 μL of a bacteria inoculum (1 × 108 CFU mL^−1^) was placed. Plates were done in triplicate and incubated at room temperature. The reduction in the growth of *G. cingulata* and *P. cinnamomi* was calculated after 7, 14 and 21 days using the following formula: RG = (TG − CG)/CG × 100 [63], where TG is the growth of the colony in dual cultures and CG is the growth of a control group. The reduction in the growth of *M. aurantia* was calculated after 3, 6 and 9 weeks, as it is a slow-growing fungus.

### 4.5. Plant Growth-Promoting Potential of Endophytic Bacteria

Endophytic bacteria plant growth-promoting potential was tested on the same 16 isolates used on the antagonism assays. For this purpose, the siderophores production, phosphate solubilization, ammonia and indol-3-acetic-acid (IAA) production was tested.

#### 4.5.1. Siderophores Production

Siderophore production was determined according to Almoneafy et al. [64]. Bacterial isolates were grown on LB broth (10 gL^−1^ of peptone, 5 gL^−1^ of yeast extract and 5 gL^−1^ of NaCl) for 24 h. A 10 μL drop of each culture was plated on LB medium (LB broth supplemented with 12 gL^−^1 agar) with chrome azurol S (CAS) complex [65] and incubated for 3 days at 28 °C. The siderophore production was determined by the presence of an orange halo around the colony. Plates were done in triplicate.

#### 4.5.2. Phosphate Solubilization

The ability of bacterial strains to solubilize phosphate was determined as previously described [64]. Briefly, isolates were grown for 24 h in GY medium (10 g L^−1^ glucose and 2 g L^−1^ yeast extract, supplemented after autoclaving with 50 mL of a 10% K2HPO4 solution and 100 mL of a 10% CaCl2 solution). A 10 μL drop of the bacterial suspension was then plated in GYA medium (GY medium with 15 g L^−1^ agar) and incubated for 7 days at 28 °C. The presence of a clear visible halo around colonies was indicative of phosphate solubilization by the bacterial isolate. Plates were done in triplicate.

#### 4.5.3. Ammonia Production

The production of ammonia was tested as previously described by Singh and Yadav [66]. Briefly, bacterial isolates were inoculated in peptone water medium (10 gL^−1^ of peptone and 5 gL^−1^ of NaCl) and incubated for 4 days at 30 °C. Nessler’s reagent was added to the tubes and colour development from brown to yellow indicated ammonia production. The production level of ammonia was classified using a 0–2 scale: 0 indicating no production when no colour change was observed, 1 for low production when a faint yellow colour was observed, and 2 for high levels of ammonia production and clear development of the solution colour to yellow or brown. Tubes were done in triplicate.

#### 4.5.4. Indole-3-Acetic Acid Production

The production of IAA was tested following a procedure previously described [64]. Briefly, isolates were cultured on LB medium supplemented with L-tryptophan (40 μg mL^−1^) and incubated for 48 h at 30 °C and 160 rpm. After centrifugation at 10,000× *g* for 15 min, 1 mL of the filtrate culture and 1 mL of Salkowski’s reagent (1.5 mL of FeCl3.6H2O 0.5 M solution, in 80 mL of 60% (*v*/*v*) H2SO4) were mixed and incubated at room temperature for 30 min. The presence of pink colour indicates the presence of IAA. The concentration of IAA produced by each bacterial isolate was colourimetrically quantified at 530 nm using an IAA standard curve (0–25 μg mL^−1^, y = 0.0359x − 0.0349, R2 = 0.995). Tests were done in triplicate.

### 4.6. Statistical Analysis

Venn diagrams of the RAPD profiles and bacteria species from cropped and wild plants were constructed in R software version 4.0.3 [67], using the package VennDiagram [68]. The Shannon index (H’), Simpson (D), Simpson reciprocal index (1-D’) and Pielou index (J’) were obtained to compare the diversity between populations. Diversity indexes were compared by a t-student test, whereas antagonism and IAA production data were analysed by one-way ANOVA using GraphPad Prism (v. 8.4.3 for Windows, San Diego, CA, USA), followed by a Tukey’s multiple comparison test (*p* < 0.05). 

## 5. Conclusions

Due to the economic potential and ecological importance of *A. unedo* in the Mediterranean region, knowledge about this plant’s tolerance mechanisms against biotic and abiotic stress must be considerably improved, and plant protection strategies should be planned to ameliorate plant fitness. This study is the first step towards enhancing our understanding of the microbiota of *A. unedo* and provides the first identification and characterization of its endophytic culturable bacteria. Although a slightly higher number of taxa and RAPD profiles was identified in the wild population when compared to the cultivated plants, no differences were obtained in terms of diversity indexes, which seems to indicate that, in this specific case, microbiota diversity is not compromised by agricultural practices. Among the several species isolated and identified, a *Bacillus cereus* strain (Au06) proved its efficiency by antagonizing three plant pathogens, *Glomerella cingulata*, *Phytophthora cinnamomi* and *Mycosphaerella aurantia*, revealing its ecological importance and potential as a biological control agent. However, this research has raised many questions in need of further investigation, such as the contribution of horizontal and vertical transmission of microorganisms into strawberry trees. The non-culturable microbiome is currently under investigation and future research will explore the potential of *B. cereus* (Au06) as a biocontrol agent.

## Figures and Tables

**Figure 1 plants-10-01569-f001:**
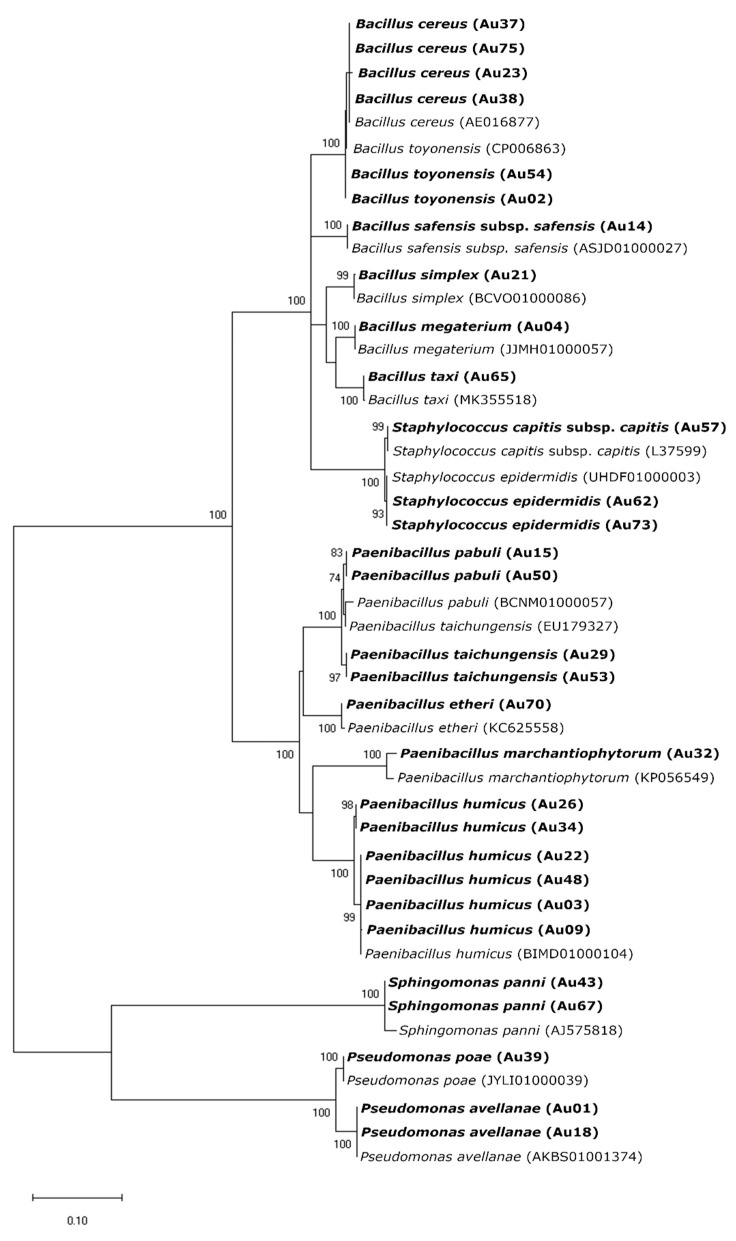
Maximum Likelihood (ML) phylogenetic tree based on 16S rRNA sequences of endophytic bacteria and sequences obtained from the EZBioCloud database. The ML tree was constructed using the Kimura 2 parameter model (K2) and gamma-distributed (+G) with invariant sites (+I) (=K2+G+I). All positions containing gaps and missing data were included for analysis. Clade supports were calculated based on 1000 replicate runs. Accession numbers (GeneBank) of the sequences belonging to the endophytic bacteria isolated in this study and the reference sequences from EZBioCloud are listed in Appendix A.

**Figure 2 plants-10-01569-f002:**
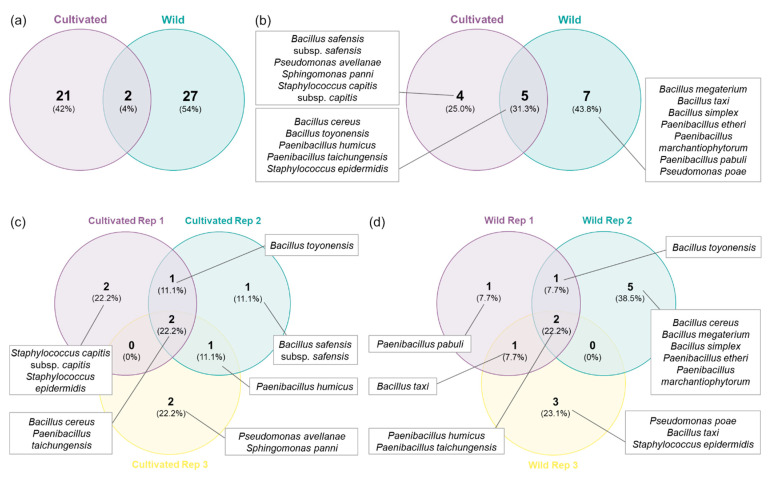
Venn diagrams with the number of RAPD profiles and bacteria species: (**a**) RAPD profiles from orchard and wild plants; (**b**) endophytic bacteria species from orchard and wild plants; (**c**) endophytic bacteria species from cultivated plant replicates; (**d**) endophytic bacteria species from wild plant replicates.

**Figure 3 plants-10-01569-f003:**
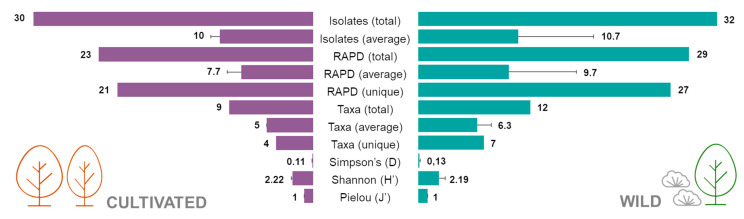
Number of isolates, RAPD profiles, taxa and diversity indexes (Simpson, Shannon and Pielou) of the endophytic bacterial isolates from cultivated and wild strawberry tree populations.

**Figure 4 plants-10-01569-f004:**
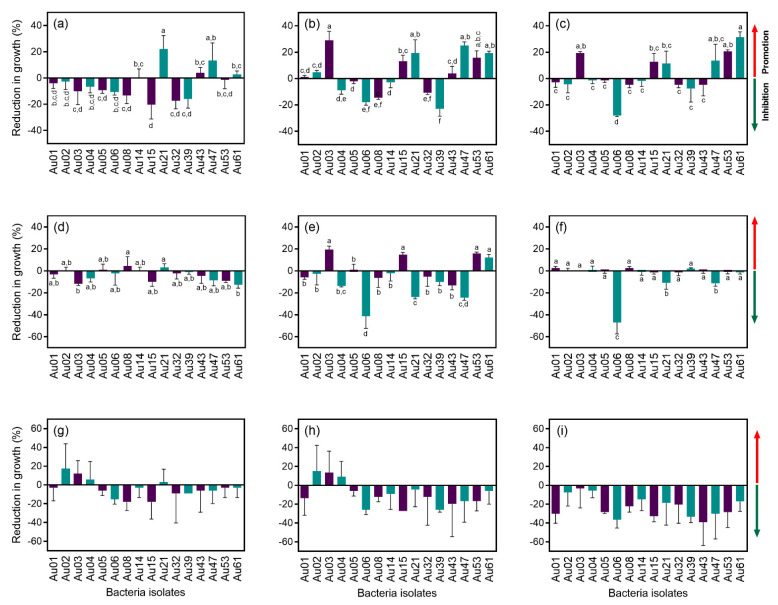
Antagonism assay between some of the endophytic bacteria isolated from *Arbutus unedo* and the plant pathogens *Glomerella cingulata*, *Mycosphaerella aurantia* and *Phytophthora cinnamomi*. Reduction in growth (%) was calculated after 1, 2 and 3 weeks for *G. cingulata* and *P. cinnamomi*, and after 3, 6 and 9 weeks for *M. aurantia*. (**a**) *G. cingulata* after 1 week; (**b**) *G. cingulata* after 2 weeks; (**c**) *G. cingulata* after 3 weeks; (**d**) *M. aurantia* after 3 weeks; (**e**) *M. aurantia* after 6 weeks; (**f**) *M. aurantia* after 9 weeks; (**g**) *P. cinnamomi* after 1 week; (**h**) *P. cinnamomi* after 2 weeks; (**i**) *Phytophthora cinnamomi* after 3 weeks. Bacteria isolates: Au01—*Pseudomonas avellanae*; Au02—*Bacillus toyonensis*; Au03—*Paenibacillus humicus*; Au04—*Bacillus megaterium*; Au05—*Bacillus toyonensis*; Au06—*Bacillus cereus*; Au08—*Bacillus toyonensis*; Au14—*Bacillus safensis*; Au15—*Paenibacillus pabuli*; Au21—*Bacillus simplex*; Au32—*Paenibacillus marchantiophytorum*; Au39—*Pseudomonas poae*; Au43—*Sphingomonas panni*; Au47—*Paenibacillus etheri*; Au53—*Paenibacillus taichungensis*; and Au61—*Bacillus taxi*. Means ± standard deviations, *n* = 3; different letters indicate significant differences between treatments (*p* ≤ 0.05).

**Figure 5 plants-10-01569-f005:**
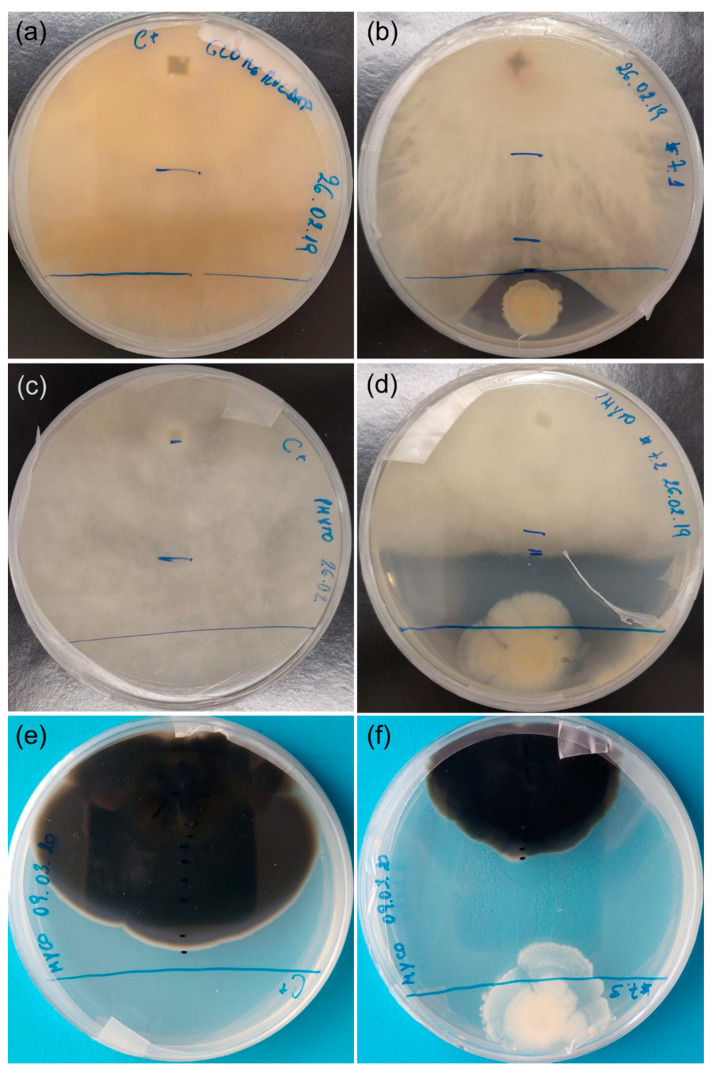
Antagonism test between Bacillus cereus (Au06) and Glomerella cingulata, Phytophthora cinnamomi and Mycosphaerella aurantia. (**a**) G. cingulata (3 weeks); (**b**) G. cingulata and B. cereus (3 weeks); (**c**) P. cinnamomi (3 weeks); (**d**) P. cinnamomi and B. cereus (3 weeks); (**e**) M. aurantia (9 weeks); (**f**) M. aurantia and B. cereus (9 weeks).

**Figure 6 plants-10-01569-f006:**
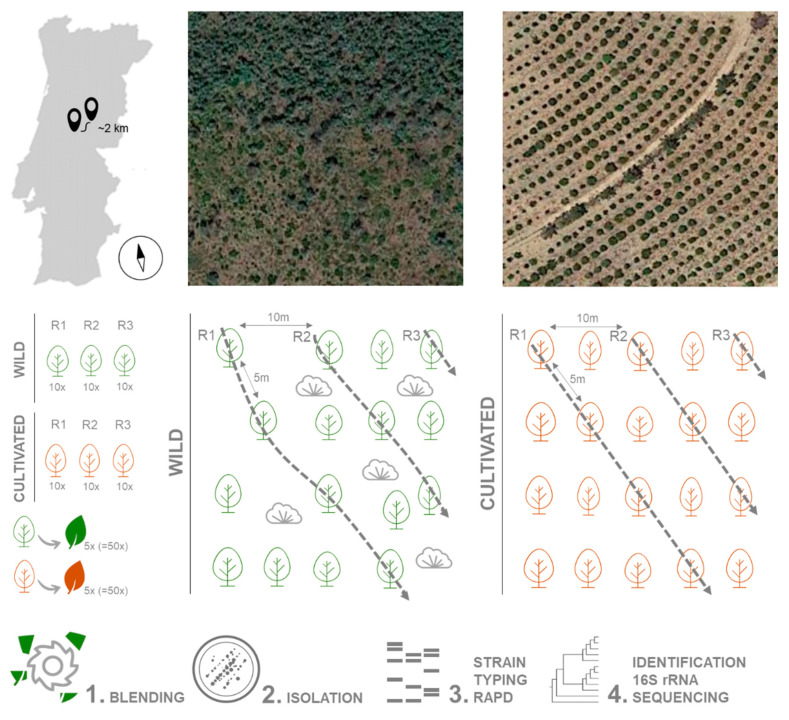
Strategies for sampling, isolation and identification of bacterial endophytes from strawberry tree leaves.

**Table 1 plants-10-01569-t001:** Bacterial endophytes isolated from *A. unedo* and some plant species where these bacteria have been identified before.

Bacteria Endophyte	Plant Species
*Bacillus cereus*	*Teucrium polium* and *Sophora alopecuroides* [17], *Solanum lycopersicum* [18], *Polygonum cuspidatum* [19], *Saccharum officinarum* [20], *Clitoria ternatea* [21], *Fragraria ananassa* and *Dyospiros kaki* [22]
*B. megaterium*	*Retama monosperma* [23], *Eucalyptus* spp. [24], *Gossypium hirsutum* and *Zea mays* [25], *S. lycopersicum* [18], *S. officinarum* [20] and *V. radiata* [26]
*B. nealsonii*	*Nicotiana attenuate* [27]
*B. safensis*	*Osmanthus fragrans* [28], *Musa* sp. [29], *P. cuspidatum* [19], *S. officinarum* [20], *Chloris virgata* [30] and *V. radiata* [26]
*B. simplex*	*P. cuspidatum* [19]
*B. taxi*	*Taxus chinensis* [31]
*B. toyonensis*	*S. lycopersicum* [18]
*Paenibacillus etheri*	-
*P. humicus*	*Acacia* sp. [32] and *Eucalyptus* spp. [24]
*P. marcantiophytorum*	*Herbertus sendtneri* [33]
*P. pabuli*	*P. cuspidatum* [19], *S. officinarum* [20] and *V. radiata* [26]
*P. taichungensis*	*V. radiata* [26]
*P. poae*	*Vitis vinifera* [34]
*Sphingomonas. panni*	*Musa* sp. [29]
*Staphylococcus capitis* subsp. *capitis*	*G. hirsutum* and *Z. mays* [25]
*S. epidermidis*	*S. lycopersicum* [35] and *Musa* sp. [29]
*P. avellanae*	*Corylus avellane* [36,37]

**Table 2 plants-10-01569-t002:** The growth-promoting potential of the isolated endophytic bacteria based on siderophore, ammonia and IAA production and phosphate solubilization. + indicates the production of IAA, siderophores and phosphate solubilization. Ammonia production: 0—no production; 1—low production; 2—high production. IAA concentration values are the means ± standard deviation of three independent replicates.

Isolate	Siderophores Production	Phosphate Solubilization	Ammonia Production	IAA Production	IAA µg mL^−1^
Au01 (*Pseudomonas avellanae*)	+	-	2	+	10.98 ± 2.44
Au02 (*Bacillus toyonensis*)	-	-	2	-	2.57 ± 1.81
Au03 (*Paenibacillus humicus*)	-	-	0	-	1.90 ± 1.81
Au04 (*Bacillus megaterium*)	-	+	2	+	6.22 ± 4.69
Au05 (*Bacillus toyonensis*)	-	-	2	-	3.10 ± 0.83
Au06 (*Bacillus cereus*)	+	-	2	-	1.72 ± 1.75
Au08 (*Bacillus toyonensis*)	-	-	2	-	4.40 ± 5.30
Au14 (*Bacillus safensis*)	+	-	2	-	2.21 ± 0.59
Au15 (*Paenibacillus pabuli*)	-	-	0	-	3.07 ± 1.42
Au21 (*Bacillus simplex*)	-	-	2	+	5.21 ± 4.36
Au32 (*Paenibacillus marchantiophytorum*)	-	+	1	-	0.56 ± 0.43
Au39 (*Pseudomonas poae*)	+	+	2	+	4.68 ± 2.11
Au43 (*Sphingomonas panni*)	+	-	1	+	6.17 ± 4.34
Au47 (*Paenibacillus etheri*)	-	-	0	-	0.74 ± 0.03
Au53 (*Paenibacillus taichungensis*)	-	-	0	-	2.47 ± 2.33
Au61 (*Bacillus taxi*)	-	+	0	+	3.87 ± 1.76
Control (*Pseudomonas syringae*)	+	-	2	+	8.43 ± 1.11
Control (*Escherichia coli*)		+	2	-	0
Negative Control	-	-	0	-	0

## Data Availability

Sequence data is available at https://www.ncbi.nlm.nih.gov/ (accessed on 27 July 2021), with accessions numbers MW534840 to MW534869.

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
