# Peer review of "Identification and Characterization of Arbutus unedo L. Endophytic Bacteria Isolated from Wild and Cultivated Trees for the Biological Control of Phytophthora cinnamomi"

_plants, 2021, doi:10.3390/plants10081569_

Round 1
Reviewer 1 Report
L. 77 the term "population" is not correct in this context, better to write group or accession, because in the population genetic sense two populations are two groups within a species without sexual contact...
L 319 it's not clar how the authors define healthy plant, this should be explained
I would suggest to include some more literature although this literature deals with forest trees, it adresses some interesting ideas, which might support the ideas of the authors: e.g.
Krabel, D., Morgenstern, K., & Herzog, S. (2013). Endophytes in changing environments - do we need new concepts in forest management? IForest - Biogeosciences and Forestry, 6(2), 109–112. https://doi.org/10.3832/ifor0932-006
Pirttilä, A. M., & Frank, A. C. (2018). Endophytes of Forest Trees. (A. M. Pirttilä & A. C. Frank, Eds.) (Second). Springer-Verlag. https://doi.org/10.1007/978-3-319-89833-9_3
Author Response
Dear Mr. Leo Zou
Assistant Editor of Plants,
We would like to acknowledge the valuable comments, corrections and suggestions raised by the reviewers that helped us to improve the manuscript. A detailed response to the comments addressed by the reviewers is given below, describing the changes made to the manuscript. Revisions in the text were made using "Track Changes" in Microsoft Word.
Reviewer #1
- 77 the term "population" is not correct in this context, better to write group or accession, because in the population genetic sense two populations are two groups within a species without sexual contact...
The term “population” was changed to “groups” as suggested.
L 319 it's not clar how the authors define healthy plant, this should be explained
Plants were considered healthy when no symptoms of disease were visible, such as black leaf spot caused by Alternaria spp. and others. “(i.e., without any disease symptoms)” was added to the text.
I would suggest to include some more literature although this literature deals with forest trees, it adresses some interesting ideas, which might support the ideas of the authors: e.g.
Krabel, D., Morgenstern, K., & Herzog, S. (2013). Endophytes in changing environments - do we need new concepts in forest management? IForest - Biogeosciences and Forestry, 6(2), 109–112. https://doi.org/10.3832/ifor0932-006
Pirttilä, A. M., & Frank, A. C. (2018). Endophytes of Forest Trees. (A. M. Pirttilä & A. C. Frank, Eds.) (Second). Springer-Verlag. https://doi.org/10.1007/978-3-319-89833-9_3
Three additional references were added to support our statements:
- Witzell, J.; Martín, J.A. Endophytes and Forest Health. In Endophytes of Forest Trees; Pirttila, A., Frank, A., Eds.; Springer, Cham, 2018; Vol. 86, pp. 261–282.
- Krabel, D.; Morgenstern, K.; Herzog, S. Endophytes in changing environments - do we need new concepts in forest management? iForest 2013, 6, 112, doi:10.3832/IFOR0932-006.
- Kubiak, K.; Wrzosek, M.; Przemieniecki, S.; Damszel, M.; Sierota, Z. Bacteria Inhabiting Wood of Roots and Stumps in Forest and Arable Soils. In Endophytes of Forest Trees, Forestry Sciences; Pirttila, A., Frank, A., Eds.; Springer, Cham, 2018; Vol. 86, pp. 319–342.
Reviewer 2 Report
The manuscript was interesting and contained valuable information. In particular, the potential use of one isolate as a biocontrol agent.
I have a few comments/edits:
How far was the wild population from the orchard? You gave GPS coordinates but it would be good to know this information.
In the results sections, which comes before the Materials and methods, I suggest you write out the names of the genera of bacteria isolated. Also, indicate what media was used and the temperatures at which the plates were incubated. It will just make easier reading.
I would exclude Table 1 as the results are reflected in the phylogenetic tree drawn. This table could be added as a supplementary table.
In table 3, indicate what 0, 1 and 2 means in terms of ammonia production.
Author Response
Answers to the reviewer’s questions
Dear Mr. Leo Zou
Assistant Editor of Plants,
We would like to acknowledge the valuable comments, corrections and suggestions raised by the reviewers that helped us to improve the manuscript. A detailed response to the comments addressed by the reviewers is given below, describing the changes made to the manuscript. Revisions in the text were made using "Track Changes" in Microsoft Word.
Reviewer #2
How far was the wild population from the orchard? You gave GPS coordinates but it would be good to know this information.
The distance between groups was approximately 2 km. This information was added to the text, as well as to Fig 6.
In the results sections, which comes before the Materials and methods, I suggest you write out the names of the genera of bacteria isolated. Also, indicate what media was used and the temperatures at which the plates were incubated. It will just make easier reading.
Information about the media, temperatures and pH was added to the text. Information about the genera is on the second paragraph (lines 98-99).
I would exclude Table 1 as the results are reflected in the phylogenetic tree drawn. This table could be added as a supplementary table.
Table 1 has been changed to supplementary material as suggested.
In table 3, indicate what 0, 1 and 2 means in terms of ammonia production.
The required information was added to the table caption, along with other information that helps the comprehension of the results. This section was also re-written in the M&M sections.